# Impact of Combined Macronutrient Diet on Amino Acids and Amines Plasma Levels

**DOI:** 10.3390/nu17101694

**Published:** 2025-05-16

**Authors:** Celia Chicharro, Patricia Romero-Marco, Luz M. González, Laura González-Rodríguez, Laura Mateo-Vivaracho, Eva Guillamón, Francisco Miguel-Tobal, Fernando Bandrés, Guillermo Gervasini, Ana Fernández-Araque, Zoraida Verde

**Affiliations:** 1Department of Biochemistry and Molecular Biology, Faculty of Health Sciences, University of Valladolid, Campus of Soria, 42004 Soria, Spain; lauramaria.mateo@uva.es (L.M.-V.); zoraida.verde@uva.es (Z.V.); 2Members of the Research Group of the Centro de Estudios Gregorio Marañón, Fundación Ortega-Marañón, 28010 Madrid, Spain; bandres@ucm.es; 3Department of Nursing, Faculty of Health Sciences, University of Valladolid, Campus of Soria, 42004 Soria, Spain; patricia.romero@uva.es (P.R.-M.); anamaria.fernandez@uva.es (A.F.-A.); 4GIR Pharmacogenetics, Cancer Genetics, Genetic Polymorphisms and Pharmacoepidemiology, University of Valladolid, Campus of Soria, 42004 Soria, Spain; 5Consolidated Research Unit of Castilla y León, 387, University of Valladolid, 42004 Soria, Spain; ggervasi@unex.es; 6Agrolab for Healthy Food, UVa, Unit Associated to CSIC Through INIA, C/Universidad s/n, 42004 Soria, Spain; 7Department of Medical and Surgical Therapeutics, Medical School, University of Extremadura, 06006 Badajoz, Spain; luzmariagg@unex.es (L.M.G.); lgonzrodrig@gmail.com (L.G.-R.); 8RICORS2040 Renal Research Network, 28029 Madrid, Spain; 9Centre for the Food Quality, INIA-CSIC, C/Universidad s/n, 42004 Soria, Spain; 10Department of Radiology, Rehabilitation and Physiotherapy, School of Medicine of Physical Education and Sport, Faculty of Medicine, University Complutense of Madrid, 28040 Madrid, Spain; miguelto@ucm.es; 11Biopathology-Toxicology Laboratory, Department of Legal Medicine, Psychiatry and Pathology, Faculty of Medicine, University Complutense of Madrid, 28040 Madrid, Spain; 12Institute of Molecular Pathology Biomarkers, University of Extremadura, 06006 Badajoz, Spain

**Keywords:** amino acids, diet, mass spectrometry, plasma

## Abstract

**Background:** Nutritional biomarkers are essential for assessing the impact of dietary interventions on health. Amino Acids (AA) serve as potential biomarkers due to their role in metabolic pathways, although the influence of combining macronutrients on AA metabolism remains unclear. **Objective:** This study aims to evaluate the effects of combining macronutrients (fat, protein, and fiber) on AA metabolism. **Methods:** A dietary intervention was conducted with 41 participants who were assigned to two groups: the Protein Fat (PF) group and the Protein Fat and Fiber (PFF) group. AA concentration was measured using LC-MS/MS. **Results:** Our findings indicated that combining macronutrients reduced plasma levels of AA with statistically significant differences between the two diets (*p* < 0.05 for EAA and BCAA; *p* < 0,01 for NEAA and AAA). Metabolic profile analysis showed differences between the groups, especially at 55 days (55 D) (R2X: 0.749; R2Y: 0.589; Q2: 0.582, *p* < 0.001) and 98 days (98 D) (R2X: 0.886; R2Y: 0.7; Q2: 0.605, *p* < 0.001). Five metabolites (leucine, asparagine, alanine, cysteine, and creatine) were the most influential factors in group differentiation (VIP score), with values ranging between 1.04 and 1.75. **Conclusions:** The combination of macronutrients led to a significant reduction in plasma amino acids and amines in the PFF group, with recovery post-intervention. These findings highlight the possibility that these metabolites are related to different health statuses.

## 1. Introduction

The study of the impact of food on human health is still evolving due to the complexity of physiological interactions that require analytical strategies to identify metabolites associated with specific dietary patterns [1,2,3,4].

Metabolomics studies endogenous or exogenous metabolites under specific physiological conditions or in response to different stimuli and employs small molecules (5 to 500 Da) and biological samples of various types, such as blood, urine, saliva, or cerebrospinal fluid [5,6,7]. It has been extensively utilized in scientific studies since the 1990s [6,7], opening the possibility of investigating the influence of dietary components on health and disease [8] and guiding nutritional strategies more precisely, contributing to better clinical patient management. For example, metabolomics studies have identified potential metabolites associated with different dietary habits and different types of foods, which can be linked to different types of diets, such as the Western diet, which is characterized by high intake of calories, animal-based proteins, saturated fat, high sugar, and low-fiber intakes [2,9,10,11,12].

Therefore, the metabolic profile of amino acids (AA) and amines may be important in the diagnosis of inherited metabolic diseases [13,14,15] as well as other conditions such as diabetes [16,17], chronic kidney disease [18,19], cancer [20,21,22,23,24,25], Alzheimer’s disease [26,27,28], cardiovascular disease [29,30], and even in protein malnutrition [31] or sepsis [32]. Moreover, it is also helpful in investigating the effect of the consumption of different food products or dietary patterns [33,34,35,36], the impacts of excessive protein and amino acid (AA) intake [37,38], and the effect of meals containing a combination of the three main macronutrients [39]. In addition, it is of great interest to have analytical techniques that offer optimal performance for obtaining the required information [40], such as Liquid Chromatography-Mass Spectrometry (LC-MS/MS) [41,42]. The data generated by these techniques in biological mixtures result in complex profiles that determine the metabolic system of each subject.

In this context, we designed a dietary intervention with the aim of observing the effects of the combined consumption of a mixed meal of dietary protein, fat, and fiber (PFF) versus dietary protein−fat (PF) on the plasma concentrations of AA and amines in a female population.

## 2. Materials and Methods

### 2.1. Subjects

A clinical trial was conducted on 41 cloistered nuns aged between 18 and 90 years. A post-hoc analysis was conducted to evaluate Statistical Power, achieving 95.5% post-hoc power for serine, 96.4% for histidine, and 99.8% for tryptophan, with an alpha of 0.05. The inclusion criteria were being a woman and a volunteer. The exclusion criteria were a diagnosis of dementia, cardiovascular disease, and/or difficulty swallowing. The study participants were randomly assigned using bias generated by the participants themselves and received one of the two proposed diets for 14 weeks.

### 2.2. Ethics

The study protocol was reviewed and approved by the BPC (CPMP/ICH/135/95) and the Ethics and Clinical Investigation Committee of the University Clinical Hospital of Valladolid (Spain). The participants recruited for the study were aware of all the aspects of the dietary intervention and signed an informed consent form prior to their inclusion in the project. The study was retrospectively registered under the number ISRCTN15421 https://www.isrctn.com/ISRCTN15421598 (accessed on 11 August 2023).

### 2.3. Dietary Intervention

The study participants were divided into two groups: the Protein Fat (PF) group, participants who consumed 150 g of pork belly without vegetables, along with the remaining food items from the menu for that day; and the Protein Fat and Fiber (PFF) group, participants who consumed 200 g of vegetables (choosing from cardoon or borage, rich in fiber and water without contributing to fat and protein intake) along with 150 g of pork belly (twice a week), and the remaining food from the daily menu. Pork belly, cardoon, and borage are typical foods widely consumed in the region. The proportion of protein and fat in pork belly was standardized to approximately 20 g of protein and 40 g of fat per 100 g of product. The total menu and physical activity were identical in both groups.

### 2.4. Amino Acids and Amines Analyzed

Essential amino acids (EAAs) include histidine, isoleucine, leucine, lysine, methionine, phenylalanine, threonine, tryptophan (Trp), and valine. Non-essential amino acids (NEAAs) include alanine, aspartate, asparagine, glutamine, glutamate, serine (Ser), citrulline, arginine, cysteine, glycine, proline, and tyrosine. The branched-chain essential amino acids (BCAAs) (leucine, isoleucine, and valine) and aromatic amino acids (AAAs) (tyrosine, phenylalanine, and tryptophan) are included in these groups. In addition, other related amines, namely acetylcholine, choline, creatine, asymmetric dimethylarginine, gamma-aminobutyric acid (GABA), kynurenic acid (KYNA), kynurenine (Kyn), and serotonin, were also analyzed. In this research, all AA were in the L-isoform, except for Ser, which exists in both L- and D-isoforms. The EAA/NEAA ratio was used to analyze the overall nutritional status [43,44,45], while the relative importance of the kynurenine pathway was determined using the Kyn/Trp ratio [46,47].

### 2.5. Chemicals and Reagents

All chemical reagents were purchased from Sigma-Aldrich (St. Louis, MO, USA). Water, methanol, and ACN (J.T. Baker, Innovagen SL, Madrid, Spain) used for the mobile phases are LC-MS/MS grade. The stock solutions of asparagine, aspartate, glycine, serine (300 µM); alanine, citrulline, glutamate, glutamine, histidine, kynurenic acid, kynurenine, methionine, phenylalanine, proline, threonine, tryptophan, valine (50 µM); arginine, ADMA, leucine, isoleucine, lysine, serotonin, tyrosine, acetylcholine (5 µM); and 0.2 µM isotopically labeled d4-acetylcholine were prepared in HPLC water and kept at −80 °C. A standard mixture was diluted from stocks with artificial cerebrospinal fluid (CSF), which consisted of 145 mM NaCl, 2.68 mM KCl, 1.4 mM CaCl_2_, 121 1.0 MgSO_4_, 1.55 mM Na_2_HPO_4,_ and 0.45 mM NaH_2_PO_4_, adjusted to pH 7.4 with NaOH.

### 2.6. Sample Collection, Preparation, and LC/MSMS Analysis

Venous blood samples (3 mL) were obtained in the fasting state and collected in EDTA tubes from each participant, aliquoted (500 μL), and preserved at −80 °C until analysis.

Sample preparation and calibration (Appendix A) followed the same procedure for both AA and amines, with the only difference being the starting volume: 20 µL for plasma samples and 5 µL for calibration standards [15]. Cold acetonitrile (1:4) was added to precipitate the proteins. The samples were then centrifuged for 10 min at 13,000 rpm to remove cell debris. The supernatant (20 µL) was derivatized by sequential addition of 10 µL of 100 mM Na_2_CO_3_, 10 µL of benzoyl chloride (BzCl) (2% *v*/*v* in acetonitrile), and 10 µL of the internal standard. In addition, the amount of organic compounds in the samples was controlled by adding 50 µL of pure water.

Both AA and amines were analyzed using liquid chromatography coupled with tandem mass spectrometry LC/MS/MS. This technique was based on the method described by Wong et al. [48]. BzCl-derivatized samples were analyzed by LC-MS/MS using multiple reaction monitoring (MRM) dynamics with an Agilent 6410 TQ instrument (Santa Clara, CA, USA). MRM conditions for all metabolites are listed in Appendix A. Five µL of the sample (in triplicate) was injected into a HiP-ALS automatic injection module at room temperature. An ACE Excel 2 SuperC18 column (15 cm × 2.1 mm ID, 2 µm, 90 Å, ACE Excel, Aberdeen, Scotland) maintained at 27 °C was used for the separation of metabolites. Mobile phase A consisted of 10 mM ammonium formate with 0.15% formic acid (pH 3), and mobile phase B consisted of acetonitrile. The flow was set to 0.2 µL/min and the elution gradient was as follows: 5% B, 0.0 min; 15% B, 0.01 min; 17% B, 0.5 min; 55% B, 14 min; 70% B, 14.5 min; 100% B, 18 min; 5% B; 19 min; 0% B, 24 min. An additional 10 min period of column equilibration at 0% B was required to achieve reproducible chromatograms. The pressure ranged from 117 to 254 bar. Electrospray ionization (ESI) was performed in the positive mode at 4000 V. The gas temperature was 350 °C and the flow was 11 L/min, with the nebulizer set to 15 psi.

Integration was performed automatically using Agilent MassHunter Quantitative Analysis for QQQ, version B.04.01.

### 2.7. Statistical Analysis

For the analysis of AA and amines (both individual and combined data; EAA, NEAA, BCAA, and AAA), quantitative variables were compared between groups using the Student’s t or Man−Whitney test according to the normality of the data distribution. Bonferroni correction was applied by dividing the number of differentiating metabolites (27 compounds) considered in the study by *p* = 0.05, resulting in a significance threshold of *p* < 0.002. Subsequent tests were performed to establish significant differences between the pairs. Two amino acid ratios were calculated to assess the increases and decreases in the ratios within each group using the Friedman Test after evaluating the data’s normality. Statistical analyses were performed using IBM SPSS Statistics software (version 26.0; IBM Corporation, Armonk, NY, USA). Subsequently, outliers identified using principal components analysis (PCA) were removed to construct Partial Least Squares Discriminant Analysis (PLS-DA) models using Umetrics SIMCA 18 software (v. 18.0.0.372). PLS-DA models allowed the identification of significantly altered metabolites between groups to generate the Variable Importance in the Projection (VIP) score, a metric that determines the relative importance of each variable in group separation or model prediction, revealing that values higher than 1 should be considered relevant. To consider the potential use of these compounds as differentiating metabolites to distinguish between dietary types, areas under the Receiver Operating Characteristic (ROC) curves (Area Under the Curve, AUC) were calculated using R Studio for Windows v. 4.2.3 (pROC package). AUCs for models containing traditional markers (age and BMI) were compared before and after adding differentiating metabolites (AA and amines selected in the VIP score) using the De Long test (to obtain the *p*-value comparing AUCs obtained with the two diet models).

## 3. Results

Forty-one participants were finally enrolled in the study. Due to various analytical tests, four samples were depleted: one from the baseline timepoint and two each from the 98 D and 132 D timepoints. Additionally, one sample from the latter time point was hemolyzed. The baseline characteristics of the two groups are shown in Table 1. The concentrations (µmol/L) of AA and other amines in the plasma are shown in Table 2.

### 3.1. Plasma Concentrations of Amino Acids and Amines

The results of the univariate analyses, following Bonferroni correction for multiple testing, showed that the differences between both groups were statistically significant for cysteine at baseline and for all metabolites (including amines, creatine choline, acetylcholine, ADMA, and kynurenine) 55 days after starting the dietary intervention (*p* < 0.002), except for methionine, tyrosine, and serotonin. At the third timepoint of the study (98 D), notable differences were found (*p* < 0.002) for histidine, lysine, valine, tryptophan, alanine, asparagine, glutamate, glutamine, citrulline, proline, cysteine, as well as for isoleucine, the latter being at the statistical limit (*p* < 0.001). Lastly, upon completion of the intervention (132 D), only tryptophan and serine levels showed significantly different values (*p* < 0.002) (Table 2).

### 3.2. Plasma Concentrations of EAA, NEAA, BCAA and AAA

Significant variations were identified between the two groups in all AA values, both during the dietary intervention and in the first 55 days.

In particular, from baseline to day 98, decreases in EAA, NEAA, BCAA, and AAA were observed in the two groups with statistically significant differences (*p* < 0.05 for EAA and BCAA; *p* < 0.01 for NEAA and AAA; Figure 1(A.2), Figure 1(C.2), Figure 1(B.2), and Figure 1(D.2), respectively). From baseline to 55 D, the decrease in plasma levels was consistent for both diets (*p* < 0.01 for EAA and NEAA; *p* < 0.05 for BCAA; *p* < 0.001 for AAA; Figure 1(A.3), Figure 1(B.3), Figure 1(C.3), and Figure 1(D.3), respectively). Finally, during the study, the differences were only statistically significant for EAA and NEAA (*p* < 0.05; Figure 1(A.1) and Figure 1(B.1), respectively).

### 3.3. Ratios of Amino Acids

Two standard ratios of plasma AA were calculated and plotted against different time points of the study to evaluate the increases and decreases in the ratio of these AAs within each group (Figure 2). Statistically significant differences (*p* < 0.001) were observed between the two groups at all time points, except for Kyn/Trp in the PFF group.

### 3.4. Metabolic Profiles

PCA analyses were conducted to identify outliers using Hotelling’s T2 distribution; they were removed before applying the PLS-DA model (Figure 3). Specifically, one outlier was removed from each of the time points 55 D, 98 D, and 132 D (their values fell outside the confidence ellipse, indicating that their concentrations were slightly different from the rest of the data set). The results of the metabolic profile analysis showed a difference in group distribution, which was clearer at 55 days (R2X: 0.749; R2Y: 0.589; Q2:0.582, *p* < 0.001; Figure 3B) and 98 days of dietary intervention (R2X: 0.886; R2Y: 0.7; Q2:0.605, *p* < 0.001; Figure 3C), with increasing values of R2X and Q2 indicating the consistency and robustness of the identified metabolic pattern. However, a decrease in sample separation capability was observed in the post-intervention phase (R2X = 0.568, R2Y = 0.785, Q2 =0.495, *p* = 0.001; Figure 3D).

### 3.5. Amino Acids and Amines as Differentiating Metabolites of the Intervention Diet

Finally, we evaluated the potential role of the five compounds identified through VIP score in assessing the impact of dietary interventions using ROC curve analysis. Two models were used: the first included classic clinical parameters (fat mass, lean mass, waist circumference, triglycerides, cholesterol, HDL, and LDL); the second model took into account the metabolic profile extracted from the VIP score, which was added to the aforementioned classic parameters. The blue line refers to the first model (classic clinical parameters), and the green line refers to the second model (classic clinical parameters plus the five metabolites from the VIP score from baseline to day 98).

As shown in Figure 4A, the inclusion of the five most relevant metabolites based on VIP score (leucine, asparagine, alanine, cysteine, and creatine) to the classic model (age, BMI, and waist circumference) resulted in a significant increase in the AUC compared to the PF group (from 78% to 100%), although this difference did not reach statistical significance. Regarding the FFI, the incorporation of significant metabolites from the VIP score (Figure 4B) into the classic model (age, BMI, and cholesterol) led to an increase in the AUC (from 76.7% to 100%), approaching the significance threshold.

## 4. Discussion

To our knowledge, this is the first study to explore the changes in plasma concentrations of AA and amines after consuming a specific intervention diet consisting of pork belly with and without vegetables in a Mediterranean female cohort.

It is important to note whether the plasma concentrations of AA in the subjects were within physiological analytical ranges; however, there is no consensus on these ranges due to discrepancies between analytical techniques and populations studied [49,50]. Various studies suggest that these differences may be due to factors such as dietary habits, race, geographical regions, and sample treatment methods [51,52,53,54].

At baseline, as expected, we found differences between the study groups for cysteine, glutamine, citrulline, arginine, and creatine, indicating pre-existing metabolic differences likely due to age differences [51,55,56,57]. A decrease in the rate of transamination and subsequent oxidation in the citric acid cycle may lead to reduced amino acid metabolism and increased blood amino acid levels during aging [58].

The changing patterns of plasma AA levels (EAA, NEAA, BCAA, and AAA) were consistent across both intervention groups (PF and PFF), with variations in the magnitude of change over time. High-fat diets have been shown to increase intestinal permeability from the intestinal lumen to the mucosal and submucosal layers, thereby altering the distribution and expression of tight junctions [59]. When intestinal permeability is compromised or increased, it can contribute to nutrient malabsorption. Therefore, the accumulation of fat digestion products could affect AA absorption, leading to a decrease in the first 55 days in the PF group and throughout the entire dietary intervention in the PFF group. However, the recovery experienced by the PF (starting from day 55) did not fully align with this hypothesis, which may be due to the homeostatic effect of the body to restore initial concentrations.

Postprandial studies have shown that carbohydrate content can slow gastric emptying and reduce postprandial plasma AA levels [60,61,62]. Our findings also indicate that combining dietary fiber from vegetables with fats and proteins from pork belly reduces plasma AA levels. Nevertheless, few studies have analyzed the effect of fat intake on the retention of AA in the intestine, but evidence suggests that fat does not appear to alter postprandial plasma AA patterns, except for a decrease in citrulline levels [39]. However, as the data were collected under fasting conditions with long intervals between measurements, further research is needed.

It is important to consider the fed or fasted state when analyzing plasma AA concentrations because the flow is different. During fasting, AA moves from the muscles to the liver and kidneys, whereas in the postprandial state, it flows from the small intestine to other organs [63]. In addition, in humans, approximately 30% to 70% of BCAAs, proline, lysine, threonine, methionine, and phenylalanine, as well as nearly all dietary glutamate, glutamine, and aspartate, are metabolized in the small intestine [64,65]. Finally, we must mention the effect of insulin, which stimulates the uptake of AA into cells after an overnight fast, thereby removing it from the peripheral circulation [66,67,68].

Furthermore, the current understanding of the relationship between AA ratios and dietary composition is limited, but preclinical studies have provided relevant data. Significant differences in the EAA/NEAA and Kyn/Trp ratios were observed in the PFF throughout the study (Figure 2). Studies conducted in animal models have shown that a high-fat diet can modulate the Kyn/Trp pathway, leading to tryptophan depletion and upregulation of the kynurenine pathway [69]. The EAA/NEAA ratio reflects the balance between essential and non-essential AA, influencing protein synthesis and indicating potential nutritional deficiencies. A low EAA/NEAA ratio suggests insufficient essential AA, while an imbalance of non-essential AA can negatively impact health [44,45].

To date, studies have mainly aimed to determine whether a metabolic profile consisting of various plasma metabolites can be used to assess adherence to and metabolic effects of the Mediterranean diet, as well as its potential link to the risk of cardiovascular disease [29,30]. This study is the first to assess the plasma concentrations of AA and amines following dietary intervention; however, it has several limitations. Despite randomization in sample selection, the study resulted in an age disparity between groups, with the PFF group consisting of older women and the PF group consisting of younger women. In addition, the dietary intake of the participants was not assessed using any FFQ. Nonetheless, our study included a group of 41 women with homogeneous dietary and lifestyle habits living in an extremely controlled environment.

## 5. Concluding Remarks

In conclusion, the combination of macronutrients significantly reduced plasma levels of AA and amines throughout the intervention in the PFF group, while levels in the PF group decreased only until day 55. Post-intervention levels tended to recover in both diets. Furthermore, adding the five most relevant metabolites identified using the VIP score significantly enhanced the evaluation of dietary intervention effects compared with the conventional approach. Finally, the AA metabolome changes following dietary intervention could serve as indicators of diet-induced variations, highlighting plasma AA and amine levels as promising nutritional differentiating metabolites. Further research is required to fully understand their utility.

## Figures and Tables

**Figure 1 nutrients-17-01694-f001:**
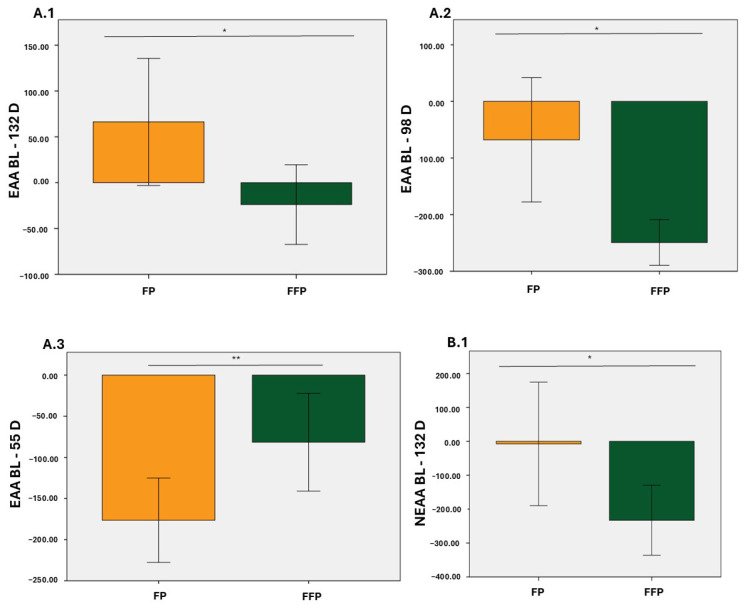
Changes in total values (µmol/L) of essential amino acids (EAA), Non-essential amino acids (NEAA), Branched-chain essential amino acids (BCAA), and aromatic amino acids (AAA). (**A.1**); Increase in EAA from BL to 132 D, (**A.2**); Increase in EAA from BL to 98 D, (**A.3**); Increase in EAA from BL to 55 D, (**B.1**); Increase in NEAA from BL to 132 D, (**B.2**); Increase in NEAA from BL to 98 D, (**B.3**); Increase in NEAA from BL to 55 D, (**C.1**); Increase in BCAA from BL to 132 D, (**C.2**); Increase in BCAA from BL to 98 D, (**C.3**); Increase in BCAA from BL to 55 D, (**D.1**); Increase in AAA from BL to 132 D, (**D.2**); Increase in AAA from BL to 98 D, (**D.3**); Increase in AAA from BL to 55 D. Orange: Protein fat (PF); Green: Protein fat and fiber (PFF). * *p* < 0.05; ** *p* < 0.01; *** *p* < 0.001.

**Figure 2 nutrients-17-01694-f002:**
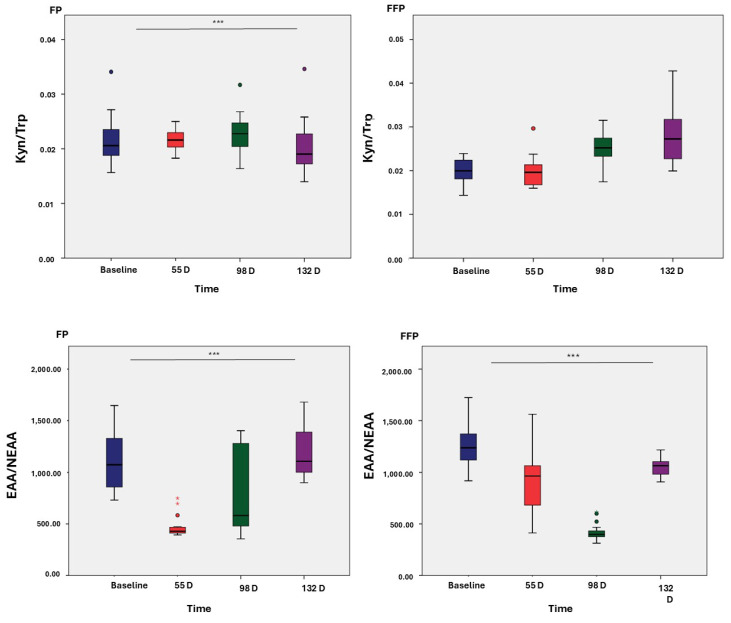
Ratios of amino acids and amines. Protein fat (PF) and protein fat and fiber (PFF). * *p* < 0.05; ** *p* < 0.01; *** *p* < 0.001.

**Figure 3 nutrients-17-01694-f003:**
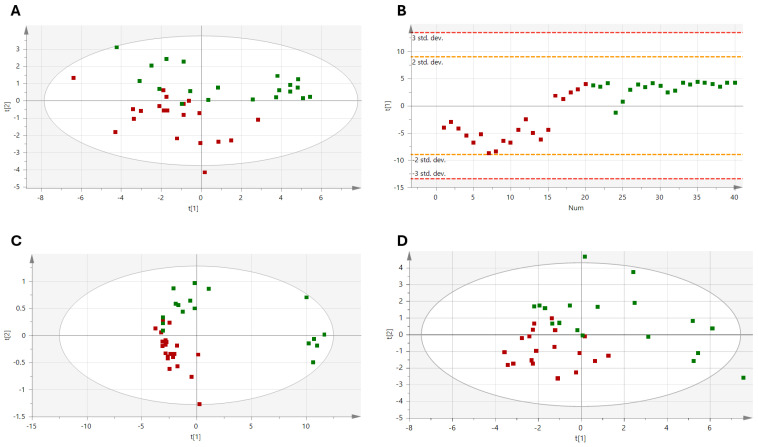
Validated PLS-DA for discrimination between the two types of dietary interventions. (**A**) Baseline; (**B**) Time 55 D; (**C**) Time 98 D; (**D**) Time 132 D. Green, PF; Red, PFF.

**Figure 4 nutrients-17-01694-f004:**
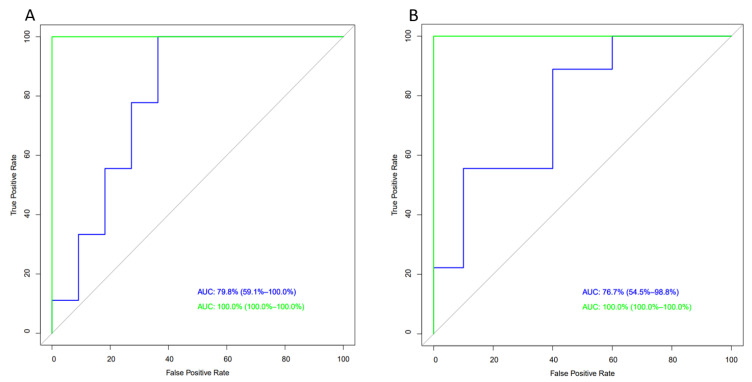
Areas under the receiver operating characteristic curves (ROC). (**A**) ROC curve for PF and (**B**) ROC curve for PFF. Blue line, classical model; green line, classical model plus the VIP score.

**Table 1 nutrients-17-01694-t001:** Baseline characteristics of the participants.

Type of Dietary Intervention							
Variables	PF			PFF			
	*n*	Mean	SD	*n*	Mean	SD	*p*-Value
AGE (years)	20	31.10	10.37	20	66.25	14.68	<0.001
BMI	20	24.39	2.45	21	28.03	4.93	<0.005
FAT (%)	20	29.06	4.30	19	33.47	7.46	<0.029
FFM (%)	20	70.95	4.30	19	66.52	7.44	<0.028
WC (cm)	20	73.38	4.75	20	85.25	13.71	<0.001
HR (bpm)	20	67.10	6.36	20	75.30	13.03	<0.016
SBP (mmHg)	20	113.50	11.83	20	118.90	14.80	<0.210
DBP (mmHg)	20	70.00	7.36	20	72.65	9.14	<0.319
CHOL (mg/dL)	20	178.67	24.26	21	206.99	27.67	<0.001
HDL (mg/dL)	20	64.68	11.58	21	66.13	9.14	<0.657
LDL (mg/dL)	20	103.85	26.42	21	124.52	30.29	<0.025
TG (mg/dL)	20	53.27	11.43	21	81.86	18.74	<0.001

Data are presented as mean  ±  SD; *p*-values were calculated using Student’s *t*-test. BMI, body mass index; CHOL, total cholesterol; DBP, diastolic blood pressure; HDL, high-density lipoprotein; HR, heart rate; LDL, low-density lipoprotein; SBP, systolic blood pressure; TG, triglycerides; WC, waist circumference. Protein fat (PF) and Protein fat and fiber (PFF).

**Table 2 nutrients-17-01694-t002:** Plasma concentrations (µmol/L) of the analyzed amino acids and amines.

	BASELINE					55 D					98 D					132 D				
	Type of Dietary Intervention					Type of Dietary Intervention					Type of Dietary Intervention					Type of Dietary Intervention				
	PF		PFF			PF		PFF			PF		PFF			PF		PPF		
	Mean	SD	Mean	SD	*p*-Value	Mean	SD	Mean	SD	*p*-Value	Mean	SD	Mean	SD	*p*-Value	Mean	SD	Mean	SD	*p*-Value
Histidine	22.38	10.07	30.19	9.77	0.017	8.04	2.03	16.15	5.37	<0.001	36.46	33.27	8.48	2.19	<0.001	57.32	17.99	44.44	8.54	0.009
Isoleucine	42.23	6.73	45.66	4.33	0.064	30.01	2.27	39.90	8.56	<0.001	36.89	8.50	28.34	3.52	<0.001	41.33	4.14	40.20	5.17	0.463
Leucine	60.38	10.45	66.21	9.47	0.072	36.96	3.34	55.91	12.06	<0.001	43.66	12.62	34.81	5.99	0.040	57.14	8.51	50.74	6.68	0.014
Lysine	29.22	8.97	32.76	5.08	0.058	9.51	3.55	29.32	12.79	<0.001	23.76	16.96	8.40	1.43	<0.001	28.39	5.66	28.58	4.58	0.909
Methionine	4.14	11.56	1.14	0.54	0.253	0.44	0.45	1.37	1.29	0.009	1.47	1.69	0.19	0.42	0.006	4.11	1.81	3.20	0.87	0.059
Phenylalanine	50.10	13.45	55.38	10.28	0.171	32.95	6.13	45.47	10.54	<0.001	35.79	14.00	27.08	6.43	0.023	51.33	6.95	48.48	5.57	0.157
Threonine	48.03	12.27	50.08	8.02	0.535	30.49	3.27	39.15	6.56	<0.001	35.29	10.27	27.24	3.87	0.005	44.23	5.98	42.54	3.48	0.295
Valine	111.41	34.53	121.19	36.11	0.387	58.29	48.48	98.56	34.48	<0.001	88.81	37.99	47.59	12.84	<0.001	142.60	41.94	125.87	37.06	0.201
Tryptophan	57.97	8.74	62.73	9.99	0.117	42.80	6.72	62.97	13.46	<0.001	53.39	21.38	33.83	5.36	<0.001	72.80	9.95	58.85	8.06	<0.001
Alanine	189.28	55.73	190.59	49.15	0.938	62.88	11.97	140.23	47.55	<0.001	105.60	48.43	56.97	13.05	<0.001	166.04	60.06	140.33	30.03	0.107
Aspartate	19.34	7.78	19.22	5.25	0.956	1.95	0.00	11.89	6.75	<0.001	9.77	10.25	1.95	0.00	0.005	14.40	5.77	9.70	3.07	0.004
Asparagine	25.46	11.78	12.94	9.92	0.388	3.09	2.10	12.95	5.77	<0.001	14.00	11.46	2.01	1.16	<0.001	18.56	6.76	15.96	3.36	0.143
Glutamate	58.81	25.66	64.63	21.43	0.441	13.53	3.44	34.26	14.46	<0.001	31.99	23.61	10.79	0.90	<0.001	39.71	11.85	31.14	3.89	0.032
Glutamine	363.74	96.58	446.08	55.91	0.008	135.73	35.64	308.56	112.39	<0.001	280.35	155.01	127.82	28.37	<0.001	421.22	88.99	367.98	29.61	0.018
Serine	8.63	4.36	7.53	3.49	0.381	1.23	0.50	3.92	2.67	<0.001	4.96	5.05	1.12	0.00	0.005	7.44	4.55	3.54	1.45	<0.001
Citrulline	33.34	11.07	42.62	10.22	0.009	11.78	3.47	28.49	12.69	<0.001	24.83	16.55	11.72	3.59	<0.001	33.53	9.15	34.97	7.38	0.595
Proline	95.08	27.09	107.26	30.41	0.189	47.21	9.65	90.63	42.11	<0.001	67.69	24.37	40.82	12.05	<0.001	103.83	18.37	99.65	27.74	0.587
Glycine	86.68	21.12	84.13	44.32	0.817	28.47	5.91	58.33	18.31	<0.001	50.44	25.51	27.62	4.20	0.003	79.99	21.65	64.98	10.57	0.012
Arginine	17.75	8.01	25.41	5.59	0.003	3.90	0.00	13.05	7.02	<0.001	13.53	13.57	3.90	0.00	0.008	17.23	8.62	14.20	3.05	0.163
Tyrosine	19.03	3.55	18.21	7.89	0.676	8.79	2.76	13.68	9.26	0.008	20.14	15.92	8.44	3.42	0.014	24.95	4.16	29.63	7.63	0.029
Cysteine	1.23	0.06	1.31	0.05	<0.001	1.12	0.01	1.27	0.10	<0.001	1.21	0.10	1.11	0.01	<0.001	1.37	0.25	1.16	0.02	0.008
Creatine	14.95	5.66	20.47	6.57	0.005	6.29	2.49	13.72	6.03	<0.001	12.97	8.24	8.53	4.05	0.048	18.77	6.16	22.41	8.16	0.030
Choline	3.22	1.22	3.23	1.02	0.979	0.88	0.64	2.74	1.31	<0.001	1.10	1.51	0.32	0.09	0.043	0.78	0.35	0.68	0.10	0.247
Acetylcholine	0.73	0.24	0.78	0.19	0.498	0.28	0.15	0.57	0.23	<0.001	0.41	0.20	0.25	0.08	0.009	0.52	0.16	0.48	0.07	0.408
ADMA	0.33	0.13	0.41	0.12	0.058	0.05	0.03	0.21	0.11	<0.001	0.16	0.14	0.06	0.04	0.004	0.24	0.06	0.20	0.03	0.022
Kynurenine	1.22	0.12	1.25	0.20	0.584	0.92	0.11	1.22	0.22	<0.001	1.18	0.41	0.83	0.05	0.002	1.45	0.22	1.59	0.24	0.068
Serotonin	0.15	0.10	0.11	0.04	0.063	0.13	0.06	0.26	0.20	0.017	0.70	1.07	0.15	0.07	0.045	0.52	0.31	0.51	0.56	0.983
EAA	425.85	89.85	465.34	72.54	0.134	249.48	58.69	384.95	98.69	<0.001	355.52	152.72	215.96	38.88	<0.001	499.25	87.60	442.91	59.12	0.026
NEAA	918.34	249.56	1035.44	194.46	0.106	319.67	66.45	717.25	246.76	<0.001	624.50	337.52	294.27	56.77	0.001	928.25	216.25	807.69	75.21	0.032
BCAA	214.03	49.49	233.06	45.86	0.215	125.26	49.15	194.36	52.85	<0.001	169.35	58.43	110.74	21.69	0.001	241.06	52.04	216.81	46.36	0.138
AAA	127.08	19.46	136.32	22.74	0.176	84.53	12.62	122.12	21.59	<0.001	109.32	48.96	69.34	12.82	0.003	149.07	18.25	135.82	13.31	0.017

Data are presented as mean  ±  SD; the *p*-value was calculated using the Bonferroni correction, and significance was determined at *p* < 0.002. Baseline: 55 D; Time 55 D: 98 D; Time 98 D: 132 D; Time 132 D. Protein fat (PF): Protein fat and fiber (PFF). Essential amino acids (EAA), Non-essential amino acids (NEAA), Branched-chain essential amino acids (BCAA), and aromatic amino acids (AAA).

## Data Availability

The data that support the findings of this study are available upon request from the corresponding author.

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
