# Peer review of "Impact of Combined Macronutrient Diet on Amino Acids and Amines Plasma Levels"

_nutrients, 2025, doi:10.3390/nu17101694_

Round 1

Reviewer 1 Report

Comments and Suggestions for Authors

            The manuscript by Chicharro et al. reports on a study evaluating the effects of combining macronutrients (fat, protein, and fiber) on amino acid (AA) metabolism. The results highlight that combining macronutrients reduces plasma levels of AA, with statistically significant differences observed between the two diets (p<0.05 for EAA and BCAA; p<0,01 for NEAA and AAA).

In my opinion, the manuscript is interesting, although some points need to be clarified, as indicated below.

1- Paragraph 2.3: The pork belly diet provided 20 g of protein and 40 g of fat. What about the vegetable diet? It's unclear.

2- Paragraph 2.4 and 2.5: I think it's important to write amino acids in the same way every time. There is no need to write them with a capital letter. Check this throughout the manuscript.

3- Please rewrite paragraph 2.6 to make it clear that the same instrumentation was used to determine amino acids and amines.

4- (line 234-235). If the D-Ser, GABA and KYNA are not determined, eliminate them from the manuscript. (see paragraph 3.1).

5- Table 1, in English we use points for decimals, not commas!

6- Some of the determinations reported in Table 2 have a very high standard deviation (e.g. histidine, alanine, asparagine and lysine in column PF; phenylalanine, alanine and asparagine in column PFF; etc. etc…. ). Why is this? The authors must provide an explanation.

7- The two groups have a significant age (mean) difference. Did the authors highlight this and take it into account in their discussion?

Author Response

Dear Reviewer,

We thank you for the opportunity to revise our MS entitled "Impact of Combined Fat and Fiber Diet on Amino Acids and Amines" for publication in Nutrients (MDPI). We thank you for your comments, which improved the quality of the MS because changes were made in accordance with your suggestions. Enclosed you will find a point-by-point response to each of the comments and the revised MS with the changes highlighted. The answers to the reviewers are marked in yellow and, in green, we have made changes related to the repetition rate.

POINT-BY-POINT RESPONSE TO COMMENTS AND SUGGESTIONS FOR AUTHORS

Comments 1: Paragraph 2.3: The pork belly diet provided 20 g of protein and 40 g of fat. What about the vegetable diet? It's unclear.

Response 1: Thank you for your question. The vegetable-based diet provided trace amounts of protein and fat. Specifically, the subjects in this study consumed cardoon and borage, which are rich in water and fiber according to their nutritional composition. Therefore, unlike pork belly, we conclude that the vegetable diet did not contribute meaningful amounts of protein or fat. We have added this information and highlighted it in yellow.

Comments 2: Paragraph 2.4 and 2.5: I think it's important to write amino acids in the same way every time. There is no need to write them with a capital letter. Check this throughout the manuscript.

Response 2: Thank you for your observation. We have revised the manuscript to ensure consistency in the formatting of amino acid names throughout the manuscript. All amino acids are now written in lowercase, according to standard scientific writing conventions. We highlighted it in yellow. 

Comments 3: Please rewrite paragraph 2.6 to make it clear that the same instrumentation was used to determine amino acids and amines.

Response 3: Thank you for your suggestion. We have rewritten paragraph 2.6 to make it clear that both amino acids and amines were analyzed using the same LC-MS/MS instrumentation.

Comments 4: (line 234-235). If the D-Ser, GABA and KYNA are not determined, eliminate them from the manuscript. (see paragraph 3.1).

Response 4: Thank you for your observation. We agree with the reviewer’s suggestion and have removed all mentions of D-serine (D-Ser), gamma-aminobutyric acid (GABA), and kynurenic acid (KYNA) from the manuscript, as these compounds were not included in the analytical determinations of this study.

Comments 5: Table 1, in English we use points for decimals, not commas!

Response 5: Thank you for your valuable feedback. We have reviewed Table 1 and made the necessary adjustments, ensuring that points are used for decimals instead of commas. We highlighted it in yellow. 

Comments 6: Some of the determinations reported in Table 2 have a very high standard deviation (e.g. histidine, alanine, asparagine and lysine in column PF; phenylalanine, alanine and asparagine in column PFF; etc. etc…. ). Why is this? The authors must provide an explanation.

Response 6: Thank you for your insightful observation regarding the high standard deviations observed for certain amino acids in Table 2.

The high standard deviations may be attributed to biological variability in the responses of the individuals to dietary interventions, due to different physiological and non-reported pathological factors; physiological factors may include age, nutritional status, and physical activity.

When analyzing human samples, the complexity of the matrix can be noticed in such high standard deviation values; particularly in the case of amino acids that can be influenced by a range of metabolic factors, inherent to each specimen.

Additionally, variations in sampling or experimental conditions might contribute to the observed fluctuations. I will ensure that this is clearly explained in the revised manuscript.

Comments 7: The two groups have a significant age (mean) difference. Did the authors highlight this and take it into account in their discussion?

Response 7: We thank the reviewer for the comment. We are aware of the significant differences at baseline regarding the variables you mentioned, including age, BMI, waist circumference, and heart rate. The subjects in our study were from a cloistered convent, where there is a mix of younger and older nuns. Keeping in mind the small size clinical research, we developed a simple randomization without taking stratification of prognostic variables into account.

Despite a randomization by the supervisor, when data interpretation was carried out, we realized that research results were biased by age. Thus, we mainly focused on comparing intra-group evolution. We had several measurement points throughout the study (baseline, 55 days, 98 days, and 132 days), allowing us to make comparisons within the same group over time. This within-group analysis has yielded very interesting results, which contribute valuable insights into our findings.

We have added the information in methods and limitations.

Reviewer 2 Report

Comments and Suggestions for Authors

This manuscript describes the results of a dietary intervention study on the impact of plasma Amino Acids and amines. This is quite a novel and well constructed study that shows important results of the interaction of fiber with fat and protein in the diet on plasma biomarkers.

Having said that, there are a few issues that would need answering or details needing inserting before publication:

Line 46 Please change clinical trial to dietary intervention.

Section 2.3 Please provide details on the daily menu, including protein/fat/carb/fibre daily or weekly averages. Were there differences in protein intake between the two groups? Also, please provide details on the baseline diet. This is important since the results show only a transient effect of the diet on the AA levels.

Lines 185-186 10mM or 0.15%? and what is the ratio of acid to base and or pH of buffer?

Table 1 There is a flagrant bias between the two groups in terms of age of the cohorts, as well as various clinical biomarkers, including cholesterol levels. Please include a better description of how the groups were formed and a reason for these differences in section 2.1.

Figures 1. Please change "increases" to "changes" as mostvariations seen can be decreases, not increases. Also, please increase font size on y axis numbers for better readability

Figures 3: Change PFF from red square to red triangles if possible + Change figure 3B for PLS-DA graph. Why not representing ellipses per cohort to show separation?

Author Response

Dear Reviewer,

We thank you for the opportunity to revise our MS entitled "Impact of Combined Fat and Fiber Diet on Amino Acids and Amines" for publication in Nutrients (MDPI). We thank you for your comments, which improved the quality of the MS because changes were made in accordance with your suggestions. Enclosed you will find a point-by-point response to each of the comments and the revised MS with the changes highlighted. The answers to the reviewers are marked in yellow and, in green, we have made changes related to the repetition rate.

POINT-BY-POINT RESPONSE TO COMMENTS AND SUGGESTIONS FOR AUTHORS

Comments 1: Line 46 Please change clinical trial to dietary intervention.

Response 1: Thank you for pointing this out. I will revise Line 46 to change "clinical trial" to "dietary intervention," as per your recommendation. We highlighted it in yellow.

Comments 2: Section 2.3 Please provide details on the daily menu, including protein/fat/carb/fibre daily or weekly averages. Were there differences in protein intake between the two groups? Also, please provide details on the baseline diet. This is important since the results show only a transient effect of the diet on the AA levels.

Response 2: We thank the reviewer for the comment. As the study was conducted with cloistered nuns, we had limited access to detailed dietary information beyond what was already provided. The specific types of protein consumed, other than the general dietary description mentioned, were not further detailed in the data available to us. Our responses are based on the available information, and we acknowledge the limitations this imposes on dietary analysis. However, we know that they all consume the same diet, so their protein intake is standardized.

Regarding the daily menu, the dietary intervention was conducted twice a week (specifically on Tuesdays and Thursdays) for a total duration of 98 days (with the 132-day mark representing a post-intervention period). As noted in the limitations of the study, we did not collect detailed information on food frequency consumption. However, we recovered the following information regarding the daily menu: the main meals comprised pulses three days a week, fish two days, and meat two days. Desserts accompanying the main meals included seasonal fruit and yogurt on two days. Participants abstained from consuming processed foods, snacks, or cakes.

Unfortunately, we did not have detailed data on daily or weekly averages of protein, fat, carbohydrate, or fiber intake. Furthermore, while we cannot provide specific details on protein intake differences between the two groups, we acknowledge the transient effect of the dietary intervention on amino acid levels, as reflected in the results.

Comments 3: Lines 185-186 10mM or 0.15%? and what is the ratio of acid to base and or pH of buffer?

Response 3: We thank the reviewer for the question. According to the analytical method from the original reference (Wong et al., 2016); Mobile phase A consisted of 10 mM ammonium formate with 0.15% formic acid (pH 3.0) and mobile phase B, acetonitrile. This information has been clarified in the revised manuscript highlighted in yellow.

Comments 4: Table 1. There is a flagrant bias between the two groups in terms of age of the cohorts, as well as various clinical biomarkers, including cholesterol levels. Please include a better description of how the groups were formed and a reason for these differences in section 2.1.

Response 4: We thank the reviewer for the comment. We are aware of the significant differences at baseline regarding the variables you mentioned, including age, BMI, waist circumference, and heart rate. The subjects in our study were from a cloistered convent, where there is a mix of younger and older nuns. Keeping in mind the small size clinical research, we developed a simple randomization without taking stratification of prognostic variables into account.

Despite a randomization by the supervisor, data interpretation was carried out, we realized that research results were biased by age. Thus, we mainly focused on comparing intra-group evolution. We had several measurement points throughout the study (baseline, 55 days, 98 days, and 132 days), allowing us to make comparisons within the same group over time. This within-group analysis has yielded very interesting results, which we believe contributes valuable insights into our findings.

We have added the information in methods and limitations.

Comments 5: Figures 1. Please change "increases" to "changes" as mostvariations seen can be decreases, not increases. Also, please increase font size on y axis numbers for better readability

Response 5: We thank the reviewer for the helpful suggestions. We have changed the word "increases" to "changes" in Figure 1, as recommended (highlighted it in yellow).

We are aware of the importance of the resolutions of Figure 1; therefore, we have improved the quality of the graphs according to our possibilities. We have incorporated the images directly into the manuscript while also saving them in a separate folder and uploading it to the system's 'Figures, Graphics, Images' section, as the Assistant Editor (Ms. Keira Du) have indicated us.

Comments 6: Figures 3: Change PFF from red square to red triangles if possible + Change figure 3B for PLS-DA graph. Why not representing ellipses per cohort to show separation?

Response 6: We thank the reviewer for the observation regarding the colors in Figure 3. Unfortunately, the figure was generated using software to which we no longer have access, and therefore we are unable to modify the color scheme at this stage. We apologize for this limitation and appreciate your understanding.

Round 2

Reviewer 1 Report

Comments and Suggestions for Authors

In my opinion, the revised manuscript has improved. Thus, I believe it is possible to publish it if the editor agrees.

Comments on the Quality of English Language

--